# Dynamic Deformation Behaviors of the Levitation Electromagnets of High-Speed Maglev Vehicle Negotiating a Sharp Horizontal Curve

**DOI:** 10.3390/s23052785

**Published:** 2023-03-03

**Authors:** Qingsong Yu, Xiaoqing Li, Qing Shao, Tian Han, Chunfa Zhao, Feng He

**Affiliations:** 1Key Laboratory of Traffic Safety on Track, Ministry of Education, School of Traffic & Transportation Engineering, Central South University, Changsha 410075, China; 2CRRC Changchun Railway Vehicles Co., Ltd., Changchun 130062, China; 3State Key Laboratory of Traction Power, Southwest Jiaotong University, Chengdu 610031, China

**Keywords:** maglev train, levitation bogie, electromagnet module, levitation control system, curve negotiation, multi-body system dynamics, elastic deformation

## Abstract

The elastic deformation of the levitation electromagnet (LM) of the high-speed maglev vehicle brings uneven levitation gaps and displacement differences between measured gap signals and the real gap in the middle of the LM, and then reduces dynamic performances of the electromagnetic levitation unit. However, most of the published literature has paid little attention to the dynamic deformation of the LM under complex line conditions. In this paper, considering the flexibility of the LM and the levitation bogie, a rigid-flexible coupled dynamic model is established to simulate deformation behaviors of the LMs of the maglev vehicle passing through the 650 m radius horizontal curve. Simulated results indicate that the deflection deformation direction of the same LM on the front transition curve is always opposite to that on the rear transition curve. Similarly, the deflection deformation direction of a left LM on the transition curve is opposite to that of the corresponding right LM. Furthermore, deflection deformation amplitudes of the LMs in the middle of the vehicle are always very small (less than 0.2 mm). However, the deflection deformation of the LMs at both ends of the vehicle is considerably large, and the maximum deflection deformation is about 0.86 mm when the vehicle passes at the balance speed. This forms a considerable displacement disturbance for the nominal levitation gap of 10 mm. It is necessary to optimize the supporting structure of the LM at the end of the maglev train in the future.

## 1. Introduction

The electromagnetically suspended (EMS) maglev train utilizes magnetic attraction forces between the on-board electromagnets and the ferromagnetic rails to support and guide vehicles, the non-contact magnetic forces avoid mechanical impact, noise, and wear caused by the wheel-rail contact in traditional rail transportation. Furthermore, high-speed EMS maglev train is propelled by the linear synchronous motor, so it can be easily accelerated to over 500 km/h without derailment risk. However, since the open-loop electromagnetic levitation system is inherently unstable, feedback controllers must be used to regulate coil currents of the electromagnets for maintaining about a 10 mm levitation gap. In order to improve the loading capacity of the maglev vehicle, many levitation electromagnets (LMs) are continuously arranged along both undersides of the vehicle; meanwhile, the lightweight electromagnet is necessary. As for the German TR08 maglev vehicle, each levitation bogie is installed with a pair of LMs; moreover, two adjacent levitation bogies are lapped by the other pair of LMs [1,2]. Generally, each LM adopts two or three feedback controllers to obtain redundant levitation, hence the flexible aluminum frame (levitation bogie) is preferred to reduce mechanical coupling among multiple levitation control units. In a word, the LM and the levitation bogie are characterized by considerable elasticity, so their flexible deformations should be paid more attention because they can significantly affect the dynamic performances of the maglev vehicle.

However, most of the previous research on the maglev vehicle dynamic simulation simplifies or ignores the flexibility of the LM and the levitation bogie [3]. In order to assess the elastic deformation of the levitation bogie, Zhao et al. built a flexible body dynamics model of the levitation bogie using the modal synthesis method and then developed a rigid-flexible coupled dynamic model of a high-speed maglev vehicle to simulate its dynamic behaviors. Calculated results show that the torsional deformation of the levitation bogie passing through the 2260 m radius horizontal curve is considerably notable [4]. Thus, an equivalent flexible levitation bogie model regarding the longitudinal beam as a torsional spring, namely, two rigid C-shaped frames linked with the linear springs, was proposed to improve computational efficiency [5]. Using an equivalent flexible levitation bogie model, Wang et al. and Xu et al. developed a dynamic interaction model between the maglev vehicle and the flexible straight and curve girder. The accuracy and effectiveness of the model are validated by comparing the computed dynamic responses with the measurement results, then it is used to explore the influence of the flexibilities of the rails, girders, elastic supports, and curve parameters on the ride quality of the maglev vehicle [6,7,8]. Gong proposed a new structural form of separated track and beam for high-speed maglev and its dynamic characteristics under a moving train were studied. The results show that increasing the beam height, reducing the beam span, or improving the performance of the secondary suspension system of the train can upgrade the ride comfort further. However, the vehicle model only considers vertical vibration with a total of 10 degrees of freedom [9]. Chen et al. studied the dynamic effects of secondary suspension parameters on maglev vehicles under various track irregularities and proposed a secondary suspension scheme without bolster adding an anti-roll bar, which can not only simplify the structure but also meet the dynamic performance requirements, where the levitation bogie was modeled as two rigid bodies linked with an equivalent torsional spring [10,11]. Considering the torsional deformation of the levitation bogie and nonlinear characteristics of the secondary air spring, Liang et al. [12] and Luo et al. [13] investigated the dynamic performances of the maglev vehicle running on the vertical curves. Zhang et al. compared simulated results from two types of maglev vehicle dynamic models: one adopts the equivalent flexible levitation bogie while the other involves a complete flexible bogie based on the modal synthesis method. Then they pointed out that the fluctuation amplitudes of magnetic forces and levitation gaps calculated from the latter model are greater than the former [14]. Adopting a rigid-flexible coupled dynamic model considering the flexibility of the levitation bogie and the LM, He et al. investigated the influence of the bending rigidity of the LM on the dynamic performances of a maglev vehicle running on horizontal curves [15]. Li et al. analyzed static deflection deformation of the LM of a high-speed maglev vehicle under the maximum allowable loads, and then proposed a pre-camber scheme to improve the straightness of the magnetic pole assembly surface under the levitating operation condition [16]. In sum, the above-mentioned studies presented partially elastic deformation behaviors of the running parts of maglev vehicle, but they paid more attention to the elastic deformation of the levitation bogie while the flexibility of the LM was rarely considered. However, the elastic deformation of the LM, especially the deflection deformation, can create an uneven gap distribution along the length direction and considerable differences between the measured clearance signals at both ends of the LM and the real levitation gap in the middle. These issues, resulting from the elastic deformation of the LM, are unfavorable for the levitation control. However, this information cannot be obtained from the existing research, whether it is the deformation law or the deformation amplitude of the electromagnet. It is not clear whether the levitation electromagnet needs further optimization nor how to optimize it. Hence, a comprehensive understanding of the deformation behaviors of the LMs is necessary to explore the optimization direction of the LM of the high-speed maglev vehicle in the future.

In this paper, a rigid-flexible coupled dynamic model is built firstly according to the Shanghai TR08 high-speed maglev vehicle. It considers the flexibility of the levitation bogie and the LM. Subsequently, the model is used to simulate the dynamic response of maglev vehicle passing through a 650 m radius horizontal curve at three typical running speeds. Furthermore, dynamic deformation behaviors of all the LMs of the maglev vehicle are presented and discussed in detail, including the local deformation, the overall deflection deformation, the deflection amplitude distribution of all the LMs, and the change laws of the deflection deformation with the curve-passing speed. Thus, this study provides comprehensive insights into elastic deformation behaviors of the running parts of a maglev vehicle passing through the horizontal curve under various operating conditions.

## 2. Dynamic Model of High-Speed EMS Maglev Vehicle

The Shanghai maglev line adopting German TR08 maglev technology has operated successfully for over twenty years at speeds of up to 430 km/h [1], and the maximum test speed reaches 501 km/h. Upgrading the design speed from 500 km/h to 600 km/h, a high-speed maglev test train composed of five carriages was unveiled by China Railway Rolling Stock Corporation (CRRC) (Beijing, China) on July 2021 [2], which utilizes more lightweight materials, a number of new manufacturing technology, the updated communication system, etc. Nevertheless, similar running parts and levitation/guidance control algorithms are still applied to the new 600 km/h maglev vehicle. More specifically, each vehicle is equipped with four levitation bogies, as shown in Figure 1. A pair of LMs and a pair of guidance electromagnets are mounted on the levitation bogie, and two adjacent bogies are articulated by a pair of LMs and a pair of guidance or brake electromagnets. It should be noted that the 3.096 m-long standard levitation electromagnet (SLM) contains 12 magnetic poles, while the long levitation electromagnet (LLM) at both ends of maglev train includes 14 magnetic poles. The SLM is installed on two bracket arms of the levitation bogie, and the LLM is installed on three bracket arms of the levitation bogie at the end of train.

### 2.1. Flexible Body Dynamic Model of Levitation Bogie and Levitation Electromagnet

As shown in Figure 1, the levitation bogie is mainly composed of the front C-shaped frame, the rear C-shaped frame, and a long longitudinal beam. The C-shaped frame is made up of four bracket arms, two cross beams, two supports for the secondary air springs, two supports for bolsters, and four short longitudinal beams. The long longitudinal beam comprises two pieces of stiffened plank beams and two articulated elements, which mainly provide a certain bending stiffness and torsional stiffness for the levitation frame, and can transfer the longitudinal load. When the vehicle passes through the transition curve of the horizontal curve, the running part can adapt to the line change through the torsion deformation of the longitudinal beam. The top of the hanger is hinged on the outer end of bolster, and then the bottom of the hanger is hinged with the carbody. The bolster-hanger mechanism can convert lateral displacements between the carbody and the LFs into swing angles of the hangers; meanwhile, it also provides the self-balancing and self-restoring capacity of the carbody and causes the air springs to almost not bear lateral loads. Because the aluminum alloy sheets are used widely in the levitation bogie structure, the SHELL181 element is adopted to create a finite element model of levitation bogie by ANSYS 19.0 software. In particular, the 3-D geometrical model of the levitation bogie considers the fillets, the rib plates, the transition surface, etc., as shown in Figure 2. Similarly, the finite element models of the SLM and the LLM are built respectively by ANSYS software. Considering the upper magnetic yokes of the LM are supported by a U-shaped thin plate beam, the LM is regarded as a beam with a constant cross-section; thus, the BEAM188 element is employed to generate the finite element model, where vertical bending stiffness of the LM is set to 9.0 MN·m^2^ [15]. Based on the finite element models of the levitation bogie, the SLM, and the LLM, the flexible body dynamic model of the running parts of maglev vehicle can be created using the Craig–Bampton method [17]. The dynamic model of high-speed maglev vehicle had been presented in our previous research [15], thus it will not be introduced in more detail here.

Generally, a floating reference frame is set up to depict the rigid body motions of a flexible body and its elastic deformation. As for the LM, an undeformed marker is set at its geometric center as the floating reference frame. In order to analyze the local and global deformation of the LM, five measuring points marked as the C1~C5 point are set at the front end, the 1/4, 1/2, and 3/4 length, and the rear end of the LM. It should be pointed out that the simulated results of elastic deformation of all the LMs in this study are dynamic components, namely, static deformation under the initial equilibrium loads is not considered in the following numerical simulation because it is almost eliminated by the pre-camber of the LM.

### 2.2. Rigid-Flexible Coupled Dynamic Model of Maglev Vehicle

Using the flexible body dynamic model shown in Figure 2, a rigid-flexible coupled dynamic model of high-speed maglev vehicle is established by SIMPACK 2018.1 software, as shown in Figure 3. In this model, the carbody, the swing bolsters, the bolster hangers, and the guidance (brake) electromagnets are regarded as rigid bodies. The secondary air springs are considered the linear spring-damping elements, as well as the rubber-metal parts connecting the LM to the bracket arm. Furthermore, the dynamic model of the single vehicle uses four LLMs at both ends of the vehicle. The left and right LMs of the maglev vehicle are labeled as the L1~L7 LM and R1~R7 LM, respectively.

### 2.3. Dynamic Model of Levitation Control Unit

Figure 4 displays the physical architecture diagram of the levitation control system for the SLM, where a total of 12 pole coils are divided into two groups. The front and rear six pole coils use, respectively, an independent levitation controller, and two sets of sensor units including the acceleration meters and the gap sensors are mounted at both ends of the SLM. In other words, every SLM contains two independent levitation control units. As for the LLMs at both ends of train, a total of 14 pole coils are divided into three groups, each of which uses an independent levitation controller. The coils with the same color in the figure indicate that their current is the same.

The dynamic model of the independent levitation control unit involves a calculation module of magnetic force and a levitation control module. Assuming the air gap between the lower surface of the stator parts and a magnetic pole is uniform, meanwhile, nonlinear magnetization characteristics of the iron core and the stator parts are ignored, and then the magnetic attractive force acting on the magnetic pole can be calculated by:(1)$$F_{m}=\frac{\mu_{0}N^{2}A}{4}{\left(\frac{I}{c}\right)}^{2}$$
where $\mu_{0}$ represents the air permeability, *N* denotes the turn number of pole coil, *A* is the effective area of the magnetic pole, *I* is the coil current, and *c* expresses the air gap.

When Formula (1) is adopted to calculate the magnetic forces of the SLM, including 12 concentrated magnetic forces acting respectively on 12 magnetic poles, the current of the front (rear) six magnetic pole coils is the same, which is regulated independently by the front (rear) levitation controller. Generally, the levitation control system uses a double-loop controller comprised of a position loop and a current loop [18,19], as shown in Figure 5. The position loop uses the PID control algorithm to maintain the stable air gap; meanwhile, acceleration feedback is added to improve the dynamic performance of the electromagnetic levitation system. On the other hand, the coil current proportional feedback is used to decrease significantly the current delay caused by the large inductance of the pole coil, so that the coil current-voltage relationship becomes an approximate proportional component [18]. Thus, assuming the equivalent resistance of the pole coil equals 1.0 Ω, then the current control law can be described as:(2)$$I(t)=K_{p}\delta (t)+K_{i}\int_{0}^{t}\delta (t)dt+K_{d}\frac{d\delta (t)}{dt}+K_{a}a(t)$$
where *δ*(*t*) is the air gap variation and *a*(*t*) is the electromagnet acceleration. *K*_p_, *K*_i_, and *K*_d_ denote, respectively, the proportional coefficient, the integral coefficient, and the differential coefficient. *K*_a_ is the feedback coefficient of electromagnet acceleration.

Based on the magnetic force calculation formula listed in Formula (1) and the levitation control algorithm shown in Figure 5, the dynamic model of the levitation control unit is built by MATLAB R2018b/Simulink software. Similarly, the dynamic model of the guidance control unit can be created; moreover, a left guidance control unit and the corresponding right one exchanges their guidance gap signals. Furthermore, the models of the levitation and guidance control units are embedded into the rigid-flexible coupled dynamic model of maglev vehicle. So far, a complete dynamic model of maglev vehicle system has been established, as shown in Figure 3, which integrates multiple rigid bodies, multiple flexible bodies, and a lot of levitation and guidance control units. Furthermore, a similar maglev vehicle dynamic model built by our research team, which uses the equivalent flexible levitation bogie and the rigid LM, had been validated by the test results [20], so we believe that the rigid-flexible coupled dynamic model of maglev vehicle in this study is credible.

## 3. Horizontal Curve Model and Its Parameters

The sharp horizontal curve can create strong geometric constraints for maglev vehicles, and then leads to large load variations and elastic deformation of the LMs. Hence, according to China’s Standard for Design of High-speed Maglev Transit (CJJ/T310-2021) [21], a horizontal curve with a radius of 650 m is set in this study. In order to simulate the whole dynamic behavior of a maglev vehicle entering into, traveling over, and departing from the horizontal curve, the curved track model is composed of five segments, including the front and rear straight lines, the front and rear transition curves, and a circular curve. More specifically, the front and rear transition curves are symmetrical with respect to the radial axis at the midpoint of the circular curve, the length of the circular curve is 160 m, the length of the transition curve is 120 m, and the transverse slope of the circular curve is 6°. It should be pointed out that the curve track model does not take into account track irregularities, in order to clearly depict elastic deformation behaviors of the LMs caused by the sharp curve and the curve-passing speed.

The horizontal curve for high-speed EMS maglev transportation adopts the one-sine-wave type transition curve [21], the curvature and the first-order derivative of curvature can be expressed as:(3)$$k(l)=\frac{1}{R_{c}}\left[\frac{1}{L_{t}}-\frac{1}{2\pi }\sin\left(\frac{2\pi l}{L_{t}}\right)\right]$$
(4)$$\frac{dk}{dl}=\frac{1}{R_{c}L_{t}}\left[1-\cos\left(\frac{2\pi l}{L_{t}}\right)\right]$$
where *k* denotes the curvature on the position of the *l* arc length away from the starting point, *R*_c_ is the radius of the circular curve, and *L*_t_ is the length of the transition curve. Figure 6 presents the relation curves between the curvature, the curvature derivative, and the arc length.

The transition curve track twists around the route centerline; meanwhile, the transverse slope (superelevation angle) is generated to counteract the centrifugal force acting on the moving vehicle. Generally, the transverse slope increases gradually along the longitudinal direction by a function similar to Equation (3), and the changing rate of transverse slope can be calculated by:(5)$$\frac{d\theta }{dl}=\frac{\theta c}{Lt}\left[1-\cos\left(\frac{2\pi l}{Lt}\right)\right]$$
where *θ*_c_ is the constant transverse slope on the circular curve. Equation (5) indicates that the changing rate at the midpoint of the transition curve reaches the maximum value, which is 0.1 °/m (the maximum allowable limit) for the aforementioned horizontal curve with a radius of 650 m.

When the maglev vehicle runs on a circular curve with a constant transverse slope, the lateral acceleration of the carbody is approximately equal to the difference between the centrifugal acceleration and the lateral component of gravitational acceleration, called the unbalanced lateral acceleration (*a*_yu_). If *a*_yu_ is zero under the condition of a certain curve-passing speed, the speed is named the balancing speed. In order to obtain good passenger ride comfort, *a*_yu_ is specified in the range from −0.5 m/s^2^ (pointing to the inside of the curve) to 1.0 m/s^2^ (pointing to the outside of the curve) [21]. Thus, the minimum allowable speed, the balancing speed, and the maximum allowable speed for the defined horizontal curve with a radius of 650 m can be calculated from the curve parameters and the allowable unbalanced acceleration limits; they are respectively equal to 67 km/h, 93 km/h, and 130 km/h.

## 4. Deformation Behaviors of the Electromagnets Passing through a Sharp Curve

Using the rigid-flexible dynamic model of the maglev vehicle and the 650 m radius horizontal curve model, the dynamic responses of maglev vehicles are simulated to figure out the primary deformation features of the LMs under three running speed conditions of 67 km/h, 93 km/h (balancing speed), and 130 km/h. The deflection deformation of all the LMs will be discussed in more detail because they could create considerable air gap differences among the 12 or 14 magnet poles of an LM.

### 4.1. Deformation Behaviors in the Case of the Balancing Speed

When the maglev vehicle passes through the 650 m radius horizontal curve at the balancing speed (93 km/h), simulated results indicate that the elastic deformation responses of the LMs at both ends of maglev vehicle are fairly close, but they are significantly greater than those of the other LMs. Thus, the L1 and R1 LMs are selected to analyze their local deformation behaviors. Figure 7 demonstrates dynamic deformation responses at the C1~C5 points of the L1 and R1 LMs, where the positive sign denotes downward deformation. It can be found that larger deformation occurs when the maglev vehicle runs on the two transition curves; moreover, deformation peaks at the C1, C3, and C5 points of the L1 and R1 LMs are much greater than those at the C2 and C4 points. Hence, we focus on local deformation behaviors of the C1, C3, and C5 points of the LMs running on the transition curves.

Figure 7 shows that the absolute values of dynamic deformation at the C1, C3, and C5 points of the LMs increase first and then decline when the maglev vehicle runs on the transition curves; the peaks appear near the midpoints of the transition curves. Local deformation peaks at the C1 and C5 points of an LM are significantly greater than those at the C3 point of the same LM, as listed in Table 1, and the maximum deformation amplitude at both ends of the L1 and R1 LMs is 0.438 mm, while that at the C3 point is 0.268 mm. In addition, it can be also seen that the direction of the local deformation at a certain measuring point of the L1 LM is opposite to that at the same point of the R1 LM when the maglev vehicle runs on the transition curves. This phenomenon implies the two LMs incur reverse deflection deformation. Furthermore, if the deformation direction at a certain measuring point of the L1 or R1 LM is upward when the maglev vehicle runs on the front transition curve, then it would turn downward on the rear transition curve, and vice versa.

In order to analyze the deflection deformation of an LM, the vertical deformation differences between the C1, C5, and the C3 points of the LM are calculated at a time step, and the larger of them is defined as the deflection deformation value of the LM at the moment. Figure 8 demonstrates the deflection deformation responses of all the LMs when the maglev vehicle passes through the 650 m radius curve at the balancing speed, where the positive sign denotes upward deflection. It can be found that the absolute values of deflection deformation of all the LMs first increase and then decline when the maglev vehicle runs on the transition curves, which agree well with the variation curve of the first order derivative of curvature (transverse slope), as shown in Figure 6b. In particular, deflection deformation peaks of the L1, R1, L7, and R7 LMs are much greater than those of the other LMs when the maglev vehicle runs on the transition curves, it is due to the torsional motion of the maglev vehicle following the torsional curve track. Moreover, even though the maglev vehicle moves on the circular curve, deflection deformation of the L1, R1, L7, and R7 LMs is significantly greater than for the other LMs. The reason is that the large lateral offsets between the levitation bogies at both ends of the vehicle and the carriage bring increment loads of the left LMs with decrement loads of the right LMs.

Figure 8 also shows that the deflection direction of an LM running on the front transition curve is contrary to that on the rear transition curve; meanwhile, the deflection direction of a left LM is always opposite to that of the corresponding right LM. As listed in Table 2, the maximum deflection deformation of the L1, L7, R1, and R7 LMs are 0.706 mm, 0.701 mm, 0.685 mm, and 0.682 mm, respectively. In other words, when the maglev vehicle passes through the 650 m radius curve at the balancing speed, the maximum arch camber of the magnet pole face of the LMs exceeds slightly 0.7 mm, which is greater than the specified limit of 0.5 mm [16].

### 4.2. Deformation Behaviors in the Case of the Maximum Allowable Speed

Figure 9 shows the deflection deformation responses of all the LMs when the maglev vehicle passes through the horizontal curve at the speed of 130 km/h. Comparing Figure 9 with Figure 8, it can be found that the deflection response curves of the left LMs shift upward and those of the right LMs shift downward. Specifically, the upward deflection peak of the L1 (L7) LM becomes larger when the maglev vehicle moves on the front (rear) transition curve, while the downward valley of the L1 (L7) LM becomes smaller on the rear (front) transition curve. On the contrary, the upward deflection peak of the R1 (R7) LM decreases on the rear (front) transition curve, while the downward valley of the R1 (R7) LM increases on the front (rear) transition curve. Furthermore, when the running speed of the maglev vehicle increases from 93 km/h to 130 km/h, the deflection deformation of the L1, R1, L7, and R7 LMs moving on the circular curve also becomes larger. The above-mentioned changes are due to the superelevation (transversal slope) on the horizontal curve being severely deficient under the 130 km/h speed condition. Then, it makes the carbody incline toward the outside of the horizontal curve so that the loads acting on the left LMs increase while that of the right LMs decrease. Nevertheless, the deflection deformation of the L2~L6 and R2~R6 LMs are relatively small.

Table 3 lists the deflection deformation amplitudes of the LMs at both ends of a maglev vehicle in the case of 130 km/h speed. It indicates that the maximum deflection deformations of the L1, L7, R1, and R7 LMs are 0.854 mm, 0.857 mm, 0.804 mm, and 0.805 mm, respectively, which increase by 21.0%, 22.3%, 17.4%, and 18.0% comparing with those in the case of 93 km/h speed (balancing speed). Similarly, the deflection deformation of the L1 and L7 LMs on the circular curve increase from 0.145 mm to 0.435 mm, while that of the R1 and R7 LMs on the circular curve increase from 0.067 mm to 0.290 mm.

### 4.3. Deformation Behaviors in the Case of the Minimum Allowable Speed

Figure 10 displays the deflection deformation responses of all the LMs when the maglev vehicle passes through the horizontal curve at the speed of 67 km/h. Comparing Figure 10 with Figure 8, it can be observed that the deflection response curves of the left LMs shift slightly downward and those of the right LMs shift slightly upward, which are contrary to the shift trends in the case of maximum allowable speed (130 km/h). The reason is that the superelevation (transversal slope) on the horizontal curve is surplus under the 67 km/h speed condition, and then it makes the carbody incline slightly toward the inside of the horizontal curve so that the load fluctuations of the LMs are similar to those in the case of 93 km/h speed. Table 4 lists the deflection deformation amplitudes of the LMs at both ends of maglev vehicle in the case of 67 km/h speed. It indicates that the maximum deflection deformation of the L1, L7, R1, and R7 LMs are 0.645 mm, 0.640 mm, 0.692 mm, and 0.688 mm, respectively. The former two decrease by 8.6% and 8.7% while the latter two increase by 1.0% and 0.9% compared with those in the case of 93 km/h speed (balancing speed). Furthermore, the deflection deformation of the L1, L7, R1, and R7 LMs on the circular curve becomes smaller when the running speed increases from 93 km/h to 67 km/h.

### 4.4. Distribution Feature of Deflection Deformation of All the Electromagnets on the Circular Curve

Figure 11 shows the distribution curves of deflection amplitudes of all the LMs when the maglev vehicle passes through the circular curve under three running speed conditions. No matter how fast the maglev vehicle runs, the deflection amplitudes of the LMs increase gradually from the middle of the vehicle to both ends; namely, the closer an LM approaches a certain end of maglev vehicle, the greater the deflection amplitude of the LM is. When the running speed increases from 67 km/h to the 130 km/h, the amplitude distribution curve of the left seven LMs moves gradually toward the positive direction, which means that the upward deflection amplitudes of the L1 and L7 LMs become larger and larger, while deflection amplitudes of the L2~L6 LMs change from the minor downward deflections to the larger upward deflections. On the contrary, the amplitude distribution curve of the right seven LMs moves gradually towards the negative direction with the increment of the running speed of maglev vehicle; namely, the downward deflection amplitudes of the R1 and R7 LMs become larger and larger, while deflection amplitudes of the R2~R6 LMs change from the minor upward deflections to the larger downward deflections.

It can be also found from Figure 11 that deflection deformation amplitudes of the L2~L6 and R2~R6 LMs are relatively small; when the maglev vehicle runs on the 650 m radius circular curve at the allowable running speeds, they are usually less than 0.2 mm. However, the deflection deformation amplitude of L1 LM increases from 0.025 mm to 0.435 mm when the running speed increases from 67 km/h to 130 km/h, which indicates the running speed has a great influence on the deflection deformation of the LMs at the end of the maglev vehicle.

## 5. Conclusions and Discussion

Considering the flexible characteristics of the LM and the levitation bogie, a rigid-flexible coupled dynamic model is established to simulate the dynamic responses of a high-speed maglev vehicle passing the 650 m radius horizontal curve at three typical running speeds. In particular, elastic deformation behaviors of the LMs are analyzed thoroughly, and the following conclusions can be drawn from this study.

When the high-speed maglev vehicle passes through the horizontal curve at the balancing speed (93 km/h), all the LMs have larger dynamic deformation on the front and rear transition curves. Moreover, dynamic deformation peaks (valleys) of the LMs at both ends of the vehicle are much greater than those of the LMs in the middle. The maximum deflection deformation of all the LMs always appears near the midpoint of the transition curves, which is about 0.7 mm. Furthermore, the deflection deformation direction of the same LM on the front transition curve is always opposite to that on the rear transition curve, which means all the LMs incur upward and downward deflection deformation. Similarly, the deflection deformation direction of a certain left LM on the front or rear transition curve is opposite to that of the corresponding right LM.

When the running speed of the maglev vehicle increases from the balancing speed to the maximum allowable speed (130 km/h), deflection response curves of the left LMs shift upward and those of the right LMs shift downward due to the deficient superelevation on the horizontal curve. Compared with the deflection deformation of the LMs in the case of the balancing speed, the maximum deflection deformation of the LM at both ends (L1, L7, R1, and R7) of the vehicle increases by 21.0%, 22.3%, 17.4%, and 18.0%, respectively, while the deflection deformation of the LMs on the circular curve also increases significantly. In addition, when the running speed decreases from the balancing speed to the minimum allowable speed (67 km/h), deflection response curves of the left LMs shift downward and those of the right LMs shift upward due to the surplus superelevation on the horizontal curve. The maximum deflection deformation of the L1 and L7 LMs decreases by 8.6% and 8.7%, and that of the R1 and R7 LMs increases by 1.0% and 0.9% compared with those in the case of the balancing speed.

In sum, when the maglev vehicle passes through the horizontal curve at the allowable normal running speeds (67 km/h ≤ *V* ≤ 130 km/h), dynamic local deformation and overall deflection deformation of the LMs in the middle of the vehicle are very small, and they are usually less than 0.2 mm. However, the elastic deformation of the LMs at both ends (L1, R1, L7, and R7) of the vehicle is relatively large, and the maximum deflection deformation is close to 0.9 mm, which is greater than the specified limit of 0.5 mm. If more external excitations, such as track irregularities, dynamic deflection of guideway girder, and wind loads, are considered in the dynamic model of a maglev vehicle system, the deflection deformation of the LMs at both ends of the vehicle would exceed 1.0 mm, which is a considerable displacement disturbance for the nominal levitation gap of 10 mm. Hence, it is necessary to optimize the supporting structure of the LM at the end of the maglev train in the future. The purpose of any optimization is to reduce the deformation of the electromagnet. From the perspective of mechanical structure, the support structure under the magnetic pole can be designed separately for the LLM to improve the bending stiffness, including using materials with higher elastic modulus, optimizing the section shape, and increasing the wall thickness. In addition, the arrangement of the sensor can be optimized so that the current of the controller is more specific to the levitation gap of each section of the electromagnet.

## Figures and Tables

**Figure 1 sensors-23-02785-f001:**
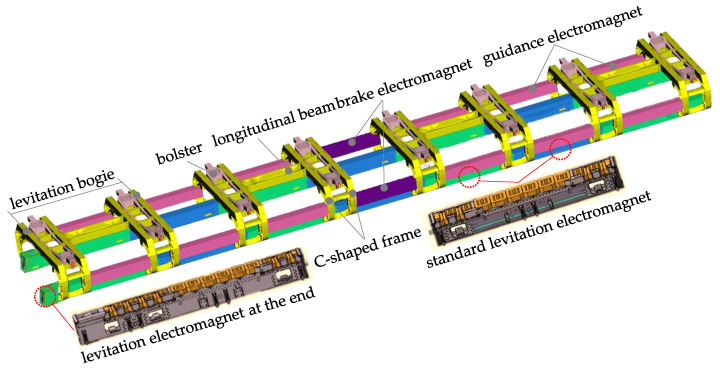
Schematic diagram of the running parts of high-speed maglev vehicle.

**Figure 2 sensors-23-02785-f002:**
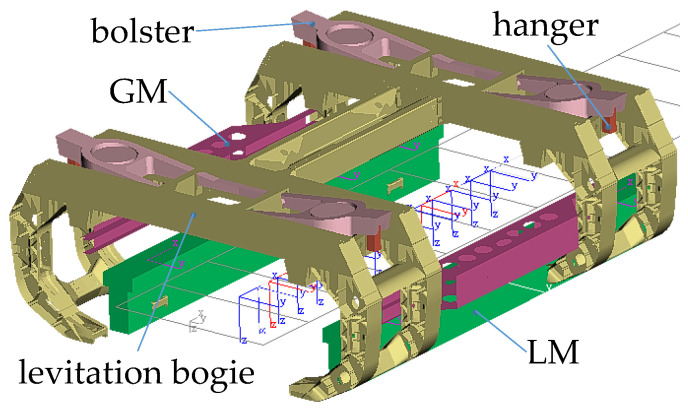
Flexible body dynamic model of the levitation bogie and the LM.

**Figure 3 sensors-23-02785-f003:**
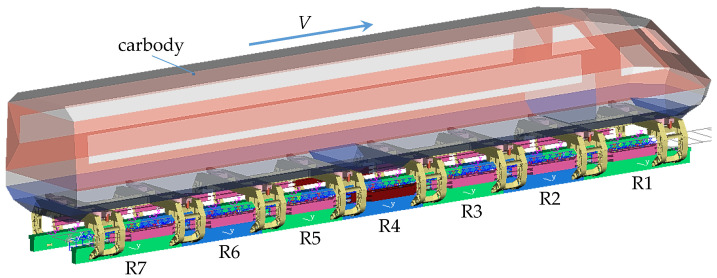
Rigid-flexible coupled dynamic model of high-speed maglev vehicle.

**Figure 4 sensors-23-02785-f004:**
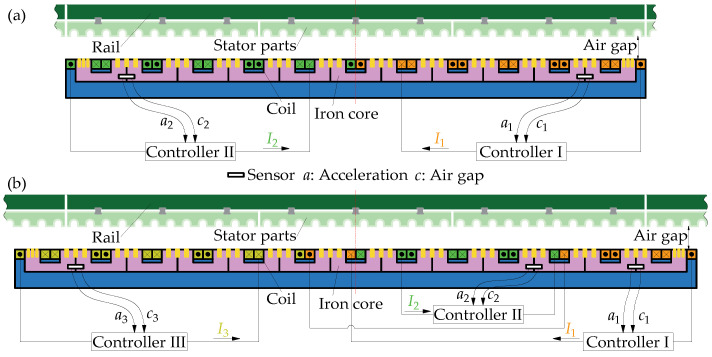
Physical architecture diagram of levitation control system: (**a**) SLM; (**b**) LLM.

**Figure 5 sensors-23-02785-f005:**
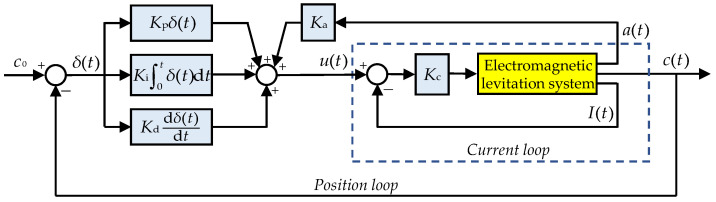
Flow chart of the levitation control system.

**Figure 6 sensors-23-02785-f006:**
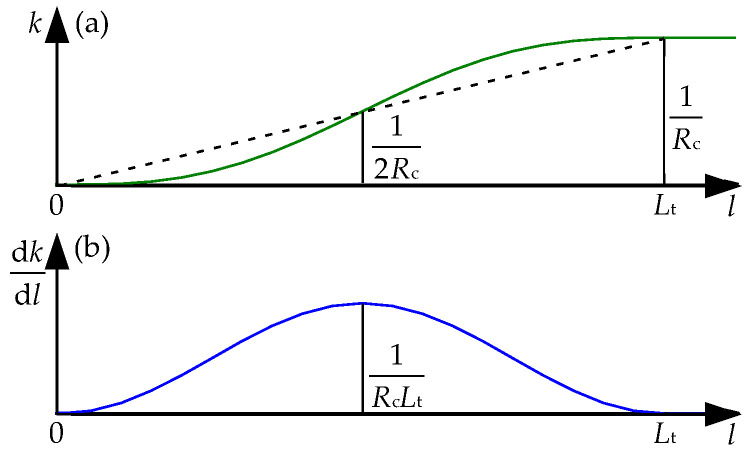
(**a**) Relation curves between the curvature and the arc length; (**b**) Relation curves between the curvature derivative and the arc length.

**Figure 7 sensors-23-02785-f007:**
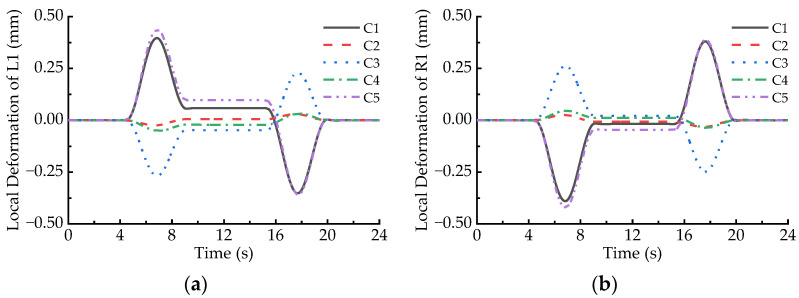
Local deformation of the LMs passing through the 650 m radius curve at the speed of 93 km/h: (**a**) L1; (**b**) R1.

**Figure 8 sensors-23-02785-f008:**
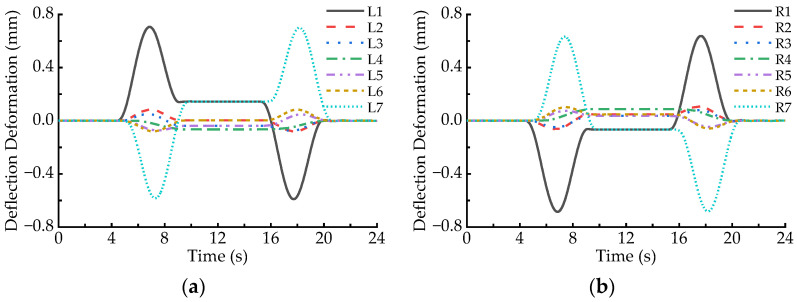
Deflection deformation of the LMs passing through the 650 m radius curve at the speed of 93 km/h: (**a**) L1~L7; (**b**) R1~R7.

**Figure 9 sensors-23-02785-f009:**
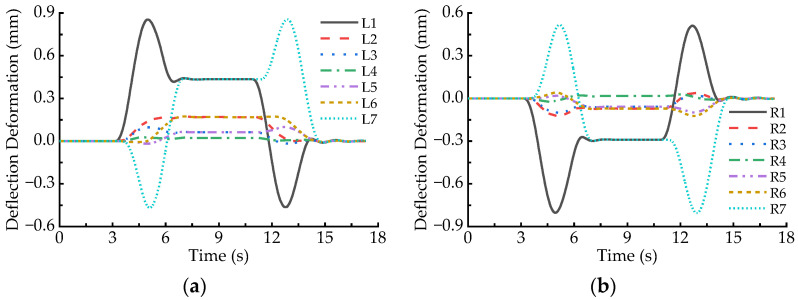
Deflection deformation of the LMs passing through the 650 m radius curve at the speed of 130 km/h: (**a**) L1~L7; (**b**) R1~R7.

**Figure 10 sensors-23-02785-f010:**
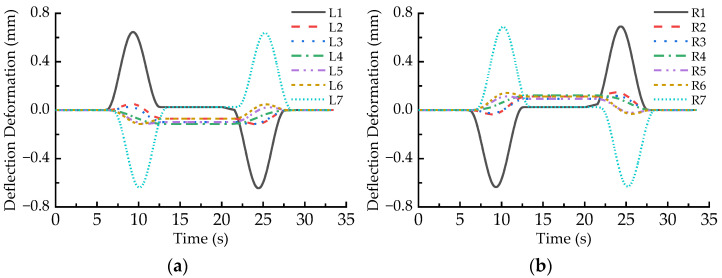
Deflection deformation of the LMs passing through the 650 m radius curve at the speed of 67 km/h: (**a**) L1~L7; (**b**) R1~R7.

**Figure 11 sensors-23-02785-f011:**
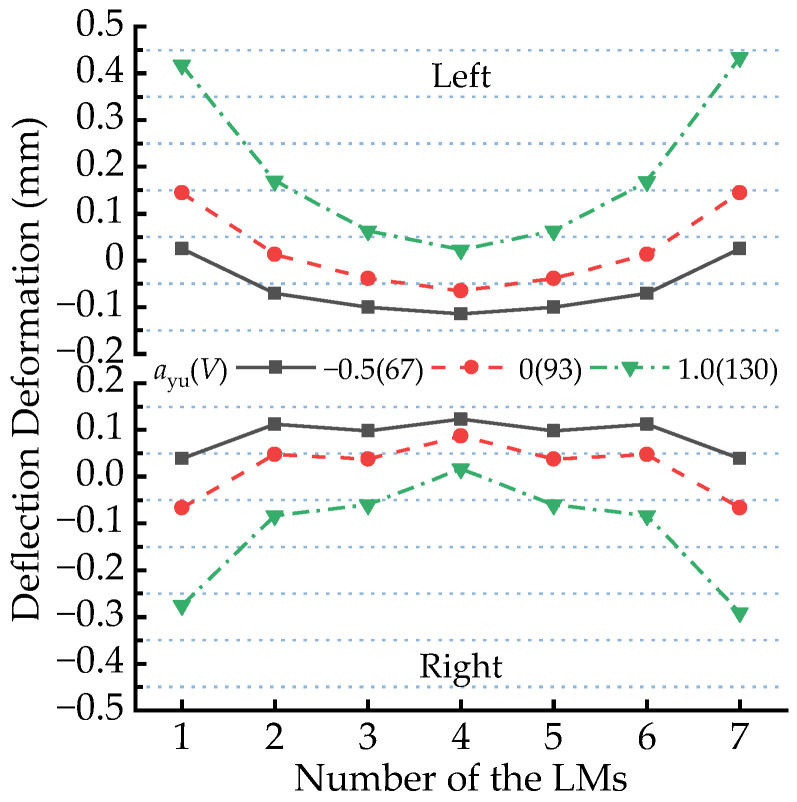
Distribution of deflection deformation amplitudes of the LMs passing through the R650 m circular curve at various running speeds.

**Table 1 sensors-23-02785-t001:** Maximum local deformation amplitudes of the L1 and R1 LMs on the 650 m radius curve (unit: mm, *V* = 93 km/h).

Position of the LM Measuring Point	L1			R1		
	C1	C3	C5	C1	C3	C5
downward deformation	0.397	0.230	0.438	0.381	0.262	0.390
upward deformation	−0.352	−0.268	−0.361	−0.390	−0.248	−0.423

**Table 2 sensors-23-02785-t002:** Maximum Deflection deformation amplitudes of the LMs on the 650 m radius curve (unit: mm, *V* = 93 km/h).

Position of the LM	L1	L7	R1	R7
On the transition curve (upward/downward)	0.706/−0.590	0.701/−0.584	0.638/−0.685	0.634/−0.682
On the circular curve	0.145	0.145	−0.067	−0.067

**Table 3 sensors-23-02785-t003:** Maximum Deflection deformation amplitudes of the LMs on the 650 m radius curve (unit: mm, *V* = 130 km/h).

Position of the LM	L1	L7	R1	R7
On the transition curve (upward/downward)	0.854/−0.464	0.857/−0.470	0.511/−0.804	0.518/−0.805
On the circular curve	0.435	0.435	−0.290	−0.290

**Table 4 sensors-23-02785-t004:** Maximum deflection deformation amplitudes of the LMs on the 650 m radius curve (unit: mm, *V* = 67 km/h).

Position of the LM	L1	L7	R1	R7
On the transition curve (upward/downward)	0.645/−0.645	0.639/−6.40	0.692/−0.636	0.688/−0.621
On the circular curve	0.025	0.025	0.025	0.025

## Data Availability

Not applicable.

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
