# Peer review of "Dynamic Deformation Behaviors of the Levitation Electromagnets of High-Speed Maglev Vehicle Negotiating a Sharp Horizontal Curve"

_sensors, 2023, doi:10.3390/s23052785_

Round 1

Reviewer 1 Report

In this paper, considering the flexibility of the levitation electromagnet and the levitation bogie, authors establish a rigid-flexible coupled dynamic model to simulate deformation behaviors of the LMs of maglev vehicle passing through the 650 m radius horizontal curve. This is a well-written paper containing interesting results which merit publication. However, a number of points need clarifying and certain statements require further justification. There are given below:

1In figure 5, authors give the flow chart of the levitation control system. However, the description of this figure is not very clear. In the figure, many arrows point to the boxes with blue background. What do these boxes represent? The authors needs to give a detailed description so that readers can understand the meaning of the figure more conveniently.

2It is noted that the manuscript needs careful editing by someone with expertise intechnical English editing paying particular attention to spelling and sentence structure so that the goals and results of the study are clear to the reader. The authors should have their work reviewed by a proper reviewing service before submission , only then can a proper review be performed. There are some sentences contain grammatical and spelling mistakes or are not complete sentences. The quality of English needs improving.

3In addition, the typesetting of the manuscript is a bit of a mess. For example, in the next line of formula 1, The position of μ0 is significantly higher than other contents of the same line. The line numbers after Figure 78 and10 are superimposed. The font of lines 308 to 314 and 318 to 331 is significantly smaller than the font of the body part. Authors need to carefully check the format and typesetting of the manuscript before resubmitting.

4、At present, there are many results for the rigid-flexible coupled dynamic model, such as “doi:10.1016/j.automatica.2021.109988” and “doi:10.1137/19M1239982”. Compared with the existing results, the main contributions of this paper need to be further strengthened.

Reviewer 2 Report

The authors present an interesting study. However, after reading the manuscript, my first impression is that it is more like a technical report than a scientific paper. The paper provides very general engineering information with no scientific value. Therefore, the contribution to science is not exposed.

The abstract is too long and should be rewritten in a more concise way. The authors are limited to the word count of the abstract. Check against the Sensors template and consider reviewing the abstract and highlighting the novelty, findings, and main conclusions.

What are the new findings of the present manuscript? What research gap did you find from previous researchers in your field? The actuality and novelty of the current problem must be clarified. Research is based on the use of specialized engineering analysis software (Ansys, Simpack). All the maths of the study is in the software used. Since the paper is submitted as an article, the scientific novelty of the paper should be made very clear. The Introduction should be extended with the most important findings. Mention it at the end of the Introduction section. It will improve the strength of the article.

Are the units of stiffness really given in line 133? Perhaps there should be division?

The text states (lines 136-137) that the detailed dynamic model is described in the publication [12]. However, for some reason, the reference list is given without internet links, which complicates the reading of the manuscript.

Perhaps that is beyond the scope of this article, but natural experiments would make the results much more reliable. Have you validated the modeling with real experiments?

In the discussion section, the authors should provide a more interesting contribution to the motivation for this study and some interesting open questions for future research. These would help to encourage the reader to take a greater interest in the topic.

Overall, my opinion is positive and I would suggest that the authors continue the work.

Reviewer 3 Report

There are a few comments for the reviewers.

1. Improve the literature review with more concerned references.

2. Where are the novel aspects of your research?

3. Your contribution must be shown in highlights.

4. You have mention two cases of speed across various parameters like 93 and 130 Km/Hr, have you compared your cases with an actual on ground model as base case?

5. How will you optimize optimize the supporting structure of the LM at the end of maglev train?

Reviewer 4 Report

Very glad to review this paper (sensors-2215656). Thanks for your waiting. This paper, considering the flexibility of the LM and the levitation bogie, established a rigid-flexible coupled dynamic model to simulate deformation behaviors of the LMs of maglev vehicle passing through the 650 m radius horizontal curve. Simulated results indicated that all the LMs occur the upward and downward deflection. However, the article also has one main problem, that is, ignoring some details, some text or picture description can be correspondingly added, so that the content of the article can be more rigorous and easier to understand. The comments below for the authors’ consideration.

Main problems:

i.          In Figure 1, some of the running parts of high-speed maglev vehicle such as bolster, longitudinal beam and C-shaped frame can be given a brief description of their functions.

ii.          With swing bolster and bolster hanger highlighted in Figure 2, a brief explanation should be given to let the reader can understand the meaning of its existence.

iii.          Below Figure 4, you had better add the physical architecture diagram of levitation control system for a LLM.

iv.          In Section 3, it is better to add a graphical illustration of the horizontal curve model, which is too abstract to be understood by words alone.

v.          In Section 4, I did not understand what for the 3 speed values are chosen (especially check L249-253 on Page 7)? In my view, three running speed conditions of 67 km/h, 93 km/h and 130 km/h are not belonging to the scope of High-speed train system. Furthermore, the scenario of “the radius of 650 m” is the only one test? Is it based on the real case? And what case should have stated.

vi.          My big concern is the manuscript is a report or an academic article? What is its contribution? An novel method or a new finding? They are unclear but very important to have the authors to think about it. 

Minor problems:

vii.          The layout of the picture in this paper is not very beautiful, and it is better to center the pictures themselves and the corresponding notes below.

viii.          The font size from line 308 to 314 and line 318 to 331 is small, so it should be modified to consistent with the main text. At the same time, the two paragraphs need to be indented in the first line.

ix.          The type size is big but sometimes small (on Page 9).

Reviewer 5 Report

The article is very well written and it's clear in their methodology, analysis and conclusions. However, I can suggest two things. First, should include the term "peak" or "maximum" deflection deformation amplitudes in Table titles to be more explicit and state which amplitudes are considered. 

In second place, For deflection deformation of LMs, I think that presenting the experimentation and results increasing (67, 93, 130 km/h) or decreasing (130, 93, 67 km/h) speeds is a better way to show the work. I think is a more obvious and logical way to analyze and derive your conclusions. In the actual state of the paper the experimentation and results were presented in speeds of 93, 130 and 67 km/h. I don't know if there is a specific reason to start the analysis at 93 km/h.

The results obtained are interesting, where the deflection deformation is stronger on the transition curves, depends (upward or downward) on the side and position  of the LM (front or rear), and speeds.

Reviewer 6 Report

COMMENTS AND FINAL OPINION:

 Author in this paper present dynamic rigid-flexible model according to the TR08 high-speed maglev vehicle. This model is used to simulate dynamic response of maglev train, which is passing through a horizontal curve with 650 m radius with different speeds (67 km/h, 93 km/h and 130 km/h). They present horizontal curve model described with parameters. With these models they figure out primary deformation features of the LMs.

 According to which criteria did you choose speeds of 67 km/h and 93 km/h?

Methodology and presented results of this research are exact and clear. References presented in the Introduction chapter, describe previous research of the problem by other authors.

 PAPER ACCEPT IN PRESENT FORM!

Round 2

Reviewer 1 Report

No further comments.

Author Response

Thank you for your review again.

Reviewer 2 Report

The authors have taken into account the comments, greatly improved the article, and answered most of the questions. I propose to accept the manuscript for publication.

Author Response

Thank you for your review again!

Reviewer 4 Report

Glad to review the manuscript (ID: sensors-2215656-revised version). Thanks for authors' revisions. The authors have addressed my concerns. A remaining comment: please make sure all references are up to date by considering possibly relevant papers that may have come out or are in press since the paper was initially submitted. Thus, the current edition is positively recommended.

Author Response

Thank you again for your suggestions and review!